# Pimozide Increases a Delayed Rectifier K^+^ Conductance in Chicken Embryo Vestibular Hair Cells

**DOI:** 10.3390/biomedicines11020488

**Published:** 2023-02-08

**Authors:** Roberta Giunta, Giulia Cheli, Paolo Spaiardi, Giancarlo Russo, Sergio Masetto

**Affiliations:** 1Department of Brain and Behavioral Sciences, University of Pavia, 27100 Pavia, Italy; 2Department of Biology and Biotechnology, University of Pavia, 27100 Pavia, Italy

**Keywords:** pimozide, hair cell, vestibular function, ionic current, K^+^ channel, patch-clamp, voltage response, chicken embryo, dizziness, balance

## Abstract

Pimozide is a conventional antipsychotic drug largely used in the therapy for schizophrenia and Tourette’s syndrome. Pimozide is assumed to inhibit synaptic transmission at the CNS by acting as a dopaminergic D_2_ receptor antagonist. Moreover, pimozide has been shown to block voltage-gated Ca^2+^ and K^+^ channels in different cells. Despite its widespread clinical use, pimozide can cause several adverse effects, including extrapyramidal symptoms and cardiac arrhythmias. Dizziness and loss of balance are among the most common side effects of pimozide. By using the patch-clamp whole-cell technique, we investigated the effect of pimozide [3 μM] on K^+^ channels expressed by chicken embryo vestibular type-II hair cells. We found that pimozide slightly blocks a transient outward rectifying A-type K^+^ current but substantially increases a delayed outward rectifying K^+^ current. The net result was a significant hyperpolarization of type-II hair cells at rest and a strong reduction of their response to depolarizing stimuli. Our findings are consistent with an inhibitory effect of pimozide on the afferent synaptic transmission by type-II hair cells. Moreover, they provide an additional key to understanding the beneficial/collateral pharmacological effects of pimozide. The finding that pimozide can act as a K^+^ channel opener provides a new perspective for the use of this drug.

## 1. Introduction

Pimozide is a conventional antipsychotic drug belonging to the diphenylbutylpiperidines class of neuroleptic agents, largely used in the therapy for schizophrenia, delusional disorders, and Tourette’s syndrome. Although it is an effective agent for treating symptoms of psychoses, it can cause several adverse effects ranging from Parkinsonian motor effects, such as akinesia and tremor, to metabolic effects including type 2 diabetes mellitus, hyperprolactinemia to cardiac arrhythmias due to QT-prolongation and altered vestibular function (reviewed in [1,2]). The mechanisms responsible for the beneficial/adverse effects of pimozide have not been completely elucidated, likely because of the many proteins targeted. Pimozide has been reported to block dopamine D_2_-receptors [3,4], voltage-gated (L− and T-type) Ca^2+^ channels [4,5,6], and different types of voltage-gated K^+^ channels [7,8] in different cell preparations. While the block of D_2_ receptors and Ca^2+^ channels will inhibit synaptic transmission, cell depolarization produced by blocking K^+^ channels may increase synaptic transmission [9].

Since dizziness and loss of balance are among the most common side effects of pimozide [2], in the present study we have investigated the effects of pimozide on the ionic currents recorded from chicken embryo vestibular type-II hair cells by using the patch-clamp whole-cell technique in combination with the slice preparation of the semicircular canals obtained at different developmental stages. Since ion channels are expressed sequentially during hair cell development [10], this preparation allows for a better dissection of their electrophysiological and pharmacological properties. The specific purpose of the present study was to define if voltage-gated K^+^ channels expressed by vestibular hair cells are affected by pimozide, and the consequences on the hair cell voltage response. Therefore, both the voltage-clamp and the current-clamp mode of the patch-clamp technique were used.

The vestibular organs of amniotes (reptiles, birds, and mammals) present two types of hair cells, named type-I and type-II hair cells—but only type-II hair cells are present in fish and amphibians. While type-II hair cells are contacted by several small (bouton) afferent nerve terminals, type-I hair cells show a unique form of innervation since their basolateral membrane is almost completely enveloped by a single giant calyx-like expansion formed by a single afferent nerve terminal [11,12,13,14,15]. Moreover, only type-I hair cells express a large sustained outward rectifying K^+^ current, termed *I*_K,L_, which activates at very negative membrane potentials (around −90 mV) and is almost completely activated at −60 mV [16,17,18]. Type-II hair cells, conversely, express a rapid transient outward rectifying K^+^ current (*I*_K,A_) (presumably of the K_V_1 and/or the K_V_4 subfamily) and a delayed sustained outward rectifying K^+^ current (K_V_2 and/or K_V_3), which activate positive to −60 mV, plus *I*_h_ (HCN) and/or the inward (anomalous) rectifying K^+^ current (*I*_K,1_) (K_ir_2 and/or K_ir_5). The present study has been focused on type-II hair cells, which represent the majority of sensory cells in avian semicircular canals and, unlike type-I hair cells, which are confined in the central region of the sensory crista epithelium, are present throughout the vestibular sensory epithelia [19]. As anticipated, the mature array of ion channels is acquired progressively during chicken embryo development according to a precisely defined schedule [10]: *I*_Ca_ (Ca_V_1) and *I*_K,v_ are acquired first (at or before embryonic day 10 (E10)), followed by *I*_K,A_, *I*_h_ and *I*_K,1_. A small Ca^2+^-dependent K^+^ current was found in a minority of type-II hair cells starting from E14, which was however always very small (<10% of the total outward K^+^ current) [10].

Our results showed that pimozide slightly blocks *I*_K,A_ while substantially increasing the amplitude of *I*_K,v_. The effect on *I*_K,v_ was consistent with a ~15 mV shift of its voltage-dependent properties towards more hyperpolarized voltages and an acceleration of the activation and inactivation kinetics. The net result of pimozide effect on *I*_K,A_ and *I*_K,v_ was an increase in the macroscopic K^+^ current at voltages near the resting membrane potential of the cell. Consistent with voltage-clamp recordings, current-clamp experiments clearly showed that the complex effect of pimozide on type-II hair cells’ K^+^ currents results in the substantial hyperpolarization of their membrane resting potential and in the reduction of the depolarization produced by positive current steps. The “opener” effect of pimozide on K,_V_ channels is a novel finding and may help elucidating some of the beneficial/adverse effects of this drug.

## 2. Materials and Methods

The chicken embryo and fetus constitute a well established model system for pharmacological studies [20]. All experiments presented here were performed on in situ vestibular type-II hair cells following chicken embryos dissection as described below.

### 2.1. Chicken Embryo Dissection

Fertilized chicken (Isa Brown) eggs were obtained by a local supplier and incubated at 37.7 °C. Experiments were performed on chicken embryos ranging from E10 to E21 (see [21] for images of the chicken inner ear during embryonic development). All animal procedures conform with the guidelines from Directive 2010/63/EU of the European Parliament on the protection of animals used for scientific purposes. The study protocol did not require the authorization of the Ministry of Health as indicated in the Legislative Decree 4 March 2014, n. 26, following the implementation of Directive (2010)/63/EU on the protection of animals used for scientific purposes. Once removed from the eggs, embryos were decapitated following brief anesthesia with 2-Bromo-2-Chloro-1,1,1-trifluoroethane (Halothane), and semicircular canals were dissected as reported below (for more details on the dissection procedure and images of the hair cell preparation, see [10,22,23]).

To preserve cell viability, heads were quickly transferred in a Petri dish filled with chilled extracellular solution (Extra_std, in mM): 145 NaCl, 3 KCl, 2 CaCl_2_, 0.6 MgCl_2_, 5.6 D-glucose, 15 HEPES; pH 7.4 with NaOH; osmolality ~310 mOsm kg^−1^.

To isolate the ampullae, surgery was continued under a stereomicroscope (Leica MZ95). The temporal bone covering the anterior and lateral ampullae was removed to expose the bony labyrinth, which was opened to expose the ampullae. The isolated ampullae were then immersed in warm agar (4%) which was rapidly solidified by adding on the surface a partially frozen extracellular solution. A small block of agar containing the ampulla was then transferred to the chamber of a microslicer DTK-1000 (DSK, Osaka, Japan). The slices of 100 µm thickness (see [10] for details) were then transferred in the recording chamber and immobilized at the bottom by mean of a nylon-mesh glued to a silver ring.

Hair cells were viewed using an upright microscope (Zeiss 2 FS plus, Oberkochen, Germany) equipped with a 10× or 67× water immersion objective.

### 2.2. Patch-Clamp Whole-Cell Recordings

Patch-clamp whole-cell experiments [24] were performed at room temperature (22 °C) by using an Axopatch 200B amplifier (Molecular Devices, USA). The analog filter was 10 kHz while sampling was 50 kHz. The patch pipettes were pulled to 2–3 MΩ tip resistance from soda glass capillaries (Hilgenberg, Germany). The patch pipette filling solution (Intra_K^+^) contained (in mM): 134 KCl, 2 MgCl_2_, 1 CaCl_2_, 11 EGTA and 10 HEPES; pH adjusted to 7.4 with KOH (osmolality: 290 mOsm kg^−1^).

Data acquisition was controlled by pClamp software using a Digidata board (Molecular Devices, San José, CA, USA). The series resistance (*R*_s_) was obtained from the average capacitive artifact elicited by voltage steps from –70 mV to –60 mV. Experiments were not considered for the analysis when *R*_s_ was > 5 MΩ.

Voltages in figures and in the text were not corrected for the liquid junction potential with the K^+^-based intracellular solution (3 mV negative inside the pipette), i.e., the nominal voltages are reported.

No leakage current subtraction was performed.

The cell resting membrane potential (*V*_rest_) was measured in current-clamp mode as the zero-current voltage.

The effect of pimozide (Tocris, UK) was tested on type-II hair cells in situ (in the slice) at different developmental stages. Pimozide was first dissolved in DMSO and then added to the extracellular solution, for a final concentration of 3 μM/L. In a few experiments, pimozide was also tested at the concentration of 0.3 μM. In preliminary experiments, we tested that the quantity of DMSO added to the extracellular solution (1%), which did not by itself affect the electrophysiological responses recorded from the hair cells. Pimozide perfusion was prolonged until a steady effect on the current or the voltage response was attained, which could require several minutes. The slice was changed after each perfusion.

### 2.3. Data Analysis and Statistical Methods

Recordings were obtained and stored for off-line analysis using pClamp 10.3 software (Molecular Devices, San Josè, CA, USA). Additional analysis was also performed by Microsoft Excel (Microsoft Corporation, Redmond, WA, USA) and OriginPro 9.0 (OriginLab, Northampton, MA, USA) software.

The equilibrium potential for K^+^ (*E*_K_) was calculated according to the Nernst equation:(1)$$E_{K}=RT/F\left(\ln\left({\left[K^{+}\right]}_{\mathrm{out}}/{\left[K^{+}\right]}_{\mathrm{in}}\right)\right)\;$$
where the subscripts ‘‘out” and ‘‘in” refer to the extracellular and intracellular solution, respectively.

The chord conductance (*G*_Ch_) was calculated for each cell at each membrane voltage as the ratio between the steady-state current and the driving force for K^+^ considering a Nernst equilibrium K^+^ voltage of −95 mV as calculated with Equation (1).

The steady-state activation curves were obtained by fitting the normalized value for *G*_Ch_ with the Boltzmann function:(2)$$G_{\mathrm{Ch}}\left(V\right)=1/(1+exp((V_{1/2}-V)/S))$$
where *G*_Ch_ (*V*) is the normalized chord conductance at voltage *V*, *V* is the membrane potential, *V*_1/2_ is the potential at which *G*_ch_ is half-maximally activated, and *S* is the voltage corresponding to an e-fold increase in *G*_ch_ (*V*).

The time-dependent inactivation of *I*_K,v_ was fitted by a single exponential function:(3)$$I\left(t\right)=Aexp\left(-t/\tau \right)+C$$
where *τ* is the inactivation time constant.

Statistical analysis was performed by Prism GraphPad 6.0 Software (San Diego, CA, USA). Following D’Agostino & Pearson and Shapiro-Wilk normality tests, statistical comparison of means was performed by paired Student’s *t*-test (two-tailed) and Wilcoxon matched-pairs signed rank test for parametric and non-parametric data, respectively.

All numerical values, degrees of freedom, and statistic values (df, t and W), in addition to the *p* values, are listed in the Appendix A. In the text, n = number of cells and mean values are quoted as means ± standard error (S.E.). In all figures, the level of statistically significant difference is as follow: * *p* ≤ 0.05; ** *p* ≤ 0.01; *** *p* ≤ 0.001; **** *p* ≤ 0.0001.

## 3. Results

### 3.1. Pimozide 3 μM on Type-II Hair Cells–Voltage-Clamp Experiments

The effect of pimozide (3 μM) was at firs*t* tested on type-II hair cells in whole-cell voltage-clamp mode. From a holding potential of −70 mV, the cells were hyperpolarized at −120 mV for 150 ms or 1000 ms and then iteratively depolarized at −40 mV, −30 mV and −20 mV for 300 ms before returning to the holding voltage.

Consistent with previous reports [10,22], in the last prenatal week (E15–E21), the conditioning step at −120 mV elicited *I*_K,1_ and/or *I*_h,_, while the subsequent depolarizing steps elicited a large outward K^+^ current consisting of a fast transient component (*I*_K,A_) followed by a slow sustained component (*I*_K,v_) (Figure 1A). Following perfusion of the extracellular solution added with pimozide, the activation and inactivation kinetics of *I*_K,v_ appeared to be accelerated, producing at −30 and −20 mV a clear second peak after that, which was due to *I*_K,A_ (Figure 1B). Moreover, the first peak increased at −40 mV and decreased at −30 and −20 mV (see also the expanded traces at the bottom of Figure 1A,B), while there was an increase in the sustained current at all the three voltages.

A possible explanation for the above results is that pimozide reduces *I*_K,A_ while it increases and accelerates *I*_K,v_. The increase in the peak current at −40 mV could in fact be due to pimozide increasing *I*_K,v_ more than it decreased *I*_K,A_. At −30 and −20 mV, conversely, the decrease in *I*_K,A_ would prevail over the increase in *I*_K,v_ on the peak current amplitude.

The voltage-dependent effect of pimozide on the peak and the sustained outward K^+^ current was further investigated in another set of cells by increasing the number of the depolarizing voltage steps up to +40 mV (Figure 2). By increasing depolarization, pimozide reduced the amplitude not only of the first peak of the outward K^+^ current, which is mainly due to *I*_K,A_, but also of the second peak and of the sustained K^+^ current, which are due to *I*_K,v_. The inhibitory effect of pimozide on *I*_K,v_ at positive voltages can be explained by the substantially accelerated inactivation of *I*_K,v_, negatively impacting the K^+^ current amplitude. The average peak and steady-state current/voltage (*I*/*V*) relations are shown in Figure 3. Note that the peak amplitude was measured in the first 17 ms of the voltage steps, as indicated by the dashed rectangle in Figure 2, and it mainly consists of *I*_K,A_ (see also the expanded traces at the bottom). The steady-state current was measured towards the end of the depolarizing step (filled squares) and consists of *I*_K,v_ since *I*_K,A_ is expected to inactivate almost completely during the 300 ms depolarizing steps [10]. The dual (increase/decrease) voltage-dependent effect of pimozide on *I*_K,v_ at most negative/positive voltages was statistically significant. For example, following pimozide perfusion, the steady-state current amplitude increased from 162.16 ± 20.05 pA to 393.90 ± 23.68 pA at −30 mV (n = 7; t = 9.381, df = 6, *p* < 0.0001 Student’s paired *t* test), while at +40 mV it decreased from 2920.74 ± 232.42 pA to 2251.01 ± 184.60 pA (n = 7; t = 5.116, df = 6, *p* = 0.0022 Student’s paired *t* test; see Appendix A).

The increase in *I*_K,v_ is a novel effect of pimozide, and it was further characterized by investigating the activation curve of *I*_K,v_. Exploiting the fact the *I*_K,A_ inactivates completely during the 300 ms depolarizing steps [10], we calculated the chord conductance at the steady-state (see Section 2) in control conditions and in the presence of pimozide. Although the amplitude of *I*_K,v_ at the steady state is clearly impacted by its inactivation, this was less pronounced at the negative voltages of interest. The data for each cell were normalized and fitted with the Boltzmann function (see Equation (2) in Section 2). The mean fits, together with the mean data superimposed (±S.E.), are shown in Figure 4. In control condition, at −70 mV *I*_K,v_ resulted to be on average only 1% active, vs. 9% in pimozide. The half-activation voltage (*V*_1/2_) was −4 ± 0.01 mV in control vs. −18 ± 0.01 mV in pimozide. Thus, pimozide shifts the activation curve of *I*_K,v_ 14 mV in the hyperpolarized direction.

Additionally, consistent with an increase in *I*_K,v_ at negative voltages, we found that the holding current at −70 mV (see the thick horizontal bar in Figure 1 for reference) changed from −17.80 ± 6.95 pA, in the control condition, to 17.87 ± 11.80 pA after pimozide administration (n = 15; E15-E21; t = 3.364, df = 14, *p* = 0.0046 Student’s paired *t* test; see Appendix A).

The leftward shift of the *I*_K,v_ activation curve might be correlated to another interesting aspect of pimozide effect, i.e., the acceleration of its activation and inactivation kinetics. To better isolate the effect of pimozide on *I*_K,v_ voltage-and time-dependent properties, in the next experiments we minimized the contribution of *I*_K,A_ to the macroscopic K^+^ current.

In a first set of experiments, we conditioned the cell membrane potential at −40 mV instead that at −120 mV to inactivate *I*_K,A_. A representative current response from a type-II hair cell is shown in Figure 5.

Vertical and horizontal red dashed lines have been drawn to better show that, at −20 mV, pimozide increased *I*_K,v_ amplitude and accelerated its activation and inactivation kinetics. For example, the time-to-peak at −20 mV decreased from 210.29 ± 30.43 ms in control condition to 106.61 ± 23.25 ms in pimozide (n = 8; t = 3.733, df = 7, *p* = 0.0073, Student’s paired *t* test). The inactivation time constant at +10 mV decreased from 1127.88 ± 222.57 ms in control condition to 759.75 ± 127.52 ms in pimozide (n = 8; t = 2.500, df = 7, *p* = 0.0410, Student’s paired *t* test).

The average values for the time-to-peak of *I*_K,v_ in control condition and in the presence of pimozide are shown in Figure 6A. Consistent with the traces shown above, the activation time course of *I*_K,v_ in the presence of pimozide accelerated at all voltages. Of notice, the time-to-peak at a given voltage in control condition, e.g., at 0 mV, was similar to that at −20 mV in pimozide, suggesting a similar shift of *I*_K,v_ voltage- and time-dependent properties in the presence of the drug.

As far as the inactivation of *I*_K,v_ is concerned, its acceleration with depolarization (Figure 5B) produced a decrease in the peak current at most depolarized voltages. Indeed, pimozide substantially accelerated *I*_K,v_ inactivation, which appeared to exceed the shift of *I*_K,v_ activation curve (Figure 6B). Although we did not investigate further this aspect, the substantial decay of *I*_K,v_ at most depolarized voltages might include an open channel block by pimozide. This has been recently shown for pimozide [3 μM] in rabbit coronary arterial smooth muscle cells, where it blocked the delayed rectifying K^+^ current in a use-dependent way, such that the decay of the current was favored by the open state of the channel as produced by strong or prolonged depolarization [25].

The average peak and steady-state *I*/*V* relations obtained by using the conditioning voltage of −40 mV are shown in Figure 7. Since *I*_K,A_ was inactivated, the peak and sustained current now consist of *I*_K,v_ only. Pimozide significantly increased the peak (n = 8; t = 6.185, df = 7, *p* = 0.0005, Student’s paired *t* test) and the steady-state current at −20 mV (n = 8; t = 3.653, df = 7, *p* = 0.0082, Student’s paired *t* test), but it showed no significant effect at −10 mV and at 0 mV, while both the peak and the steady-state currents were significantly decreased above 0 mV. These results confirm that pimozide increases or decreases *I*_K,v_ depending on the membrane voltage due to the combination of three concomitant factors: (1) the increased % of activated *I*_K,v_, (2) the acceleration of *I*_K,v_ activation kinetics, and (3) the acceleration of *I*_K,v_ inactivation kinetics.

Since *I*_K,A_ appears after *I*_K,v_ [10], in a second group of experiments type-II hair cells were chosen that did not express *I*_K,A_ (n = 4; E10–E12). A possible involvement of *I*_KCa_ can also be excluded in these experiments since *I*_KCa_ is only expressed in a minority of type-II hair cells starting from E14 [10].

A representative current response from an E10 embryo is shown in Figure 8. Note that, despite conditioning at −120 mV, the contribution of *I*_K,A_ was barely detectable. Consistent with the above experiments, in pimozide, the amplitude of the peak and the steady-state current increased at less depolarized voltages and decreased at most depolarized voltages. Moreover, the activation and inactivation kinetics accelerated (Figure 8), and the holding current was shifted upward.

### 3.2. Pimozide 3 μM on Type-II Hair Cells–Current-Clamp Experiments

Given the complex voltage-dependent action of pimozide on the macroscopic K^+^ current recorded from late embryonic developmental stages, we investigated its effect also on the voltage response recorded in current-clamp mode. One such example is shown in Figure 9. Pimozide [3 μM] produced a substantial hyperpolarization of the cell resting membrane potential and reduced the depolarization induced by the positive current steps, which is consistent with an increase in the K,v conductance at rest and at the nearby voltages. The smaller hyperpolarization produced by the negative current step of −50 pA is also most likely due to pimozide increasing the cell membrane conductance for *I*_K,v_ at rest.

The mean peak and steady-state voltage/current relations are shown in Figure 10. On average, the cell membrane resting potential shifted from −42 ± 4.48 mV to −72 ± 2.65 mV (n = 7; E15-E20; t = 7.113, df = 6, *p* = 0.0004, Student’s paired *t* test; see Appendix A).

### 3.3. Pimozide 0.3 μM on Type-II Hair Cells–Voltage-Clamp Experiments

Previous studies of the effect of pimozide, either tested at micromolar or sub-micromolar concentration, on K^+^ currents only showed a blocking action [7,8]. To better investigate this aspect, in a sample of 7 type-II hair cells, we also tested the effect of pimozide at the concentration of 0.3 μM. A representative current response is shown in Figure 11A. As also shown by the average peak and steady-state *I*–*V*s (Figure 11B), with the ten times lower pimozide concentration, the block was no longer detectable. By contrast, the increase in the peak K^+^ current was now statistically significant even at −10 mV (n = 7; E15-E21; W = 28, *p* = 0.0156, Wilcoxon matched-pairs signed rank test) and 0 mV (n = 7; E15-E21; W = 28, *p* = 0.0156, Wilcoxon matched-pairs signed rank test) (Figure 11B), and not only at −20 mV as in pimozide 3 μM (Figure 7). As far as the kinetics are concerned, a statistically significant difference was only found for the time-to-peak at −20 mV (n = 7; E15-E21; W = −24, *p* = 0.0469 Wilcoxon matched-pairs signed rank test) (Figure 11C), while no statistically significant differences were found for the inactivation time constant (Figure 11D). Thus, when compared with pimozide 3 μM (Figure 6), the effect of pimozide on *I*_K,v_ kinetics is significantly diminished.

The above results indicate an overall milder effect of pimozide 0.3 μM on *I*_K,v_ compared to 3 μM, more pronounced for its blocking effect. As a net result, the pimozide agonistic effect on *I*_K,v_ could be detected in a wider voltage range.

## 4. Discussion

The present study reveals a novel pharmacological effect of pimozide, a largely used neuroleptic drug [1], on K,v channels expressed by chicken embryo vestibular type-II hair cells. Pimozide substantially increased the amplitude of *I*_K,v_ elicited at negative membrane potentials. This effect can be explained by a leftward (toward more negative potentials) shift of *I*_K,v_ activation curve of 14 mV, such that a significantly larger fraction of K,v channels is open at rest and in the nearby voltage range. This was accompanied by an acceleration of *I*_K,v_ activation and inactivation kinetics, although an open-state block of the K,v channels by pimozide at most depolarized voltages could also be present.

The above result differs from all previous studies showing that pimozide only blocks K,v channels. More in detail, pimozide at the same concentration used here (3 μM) blocked Kv1.1 and Kv2.1 channels expressed in CHO cells [9], Kv1.5 channels in rabbit coronary arterial smooth muscle cells [25], and more generally the delayed rectifier K^+^ channels in rat hippocampal neurons [9] and (tested at 0.1 μM) in guinea pig ventricular myocytes [8]. In a sample of seven type-II hair cells, we also tested a lower concentration of pimozide (i.e., 0.3 μM) and found that the increasing effect was still present, while the blocking effect was no longer detectable.

We thought of two possible explanations for the above differences, which are not mutually exclusive possibilities: (1) a difference in K^+^ channel/subunits; (2) a difference in membrane voltages tested. As far as the latter point is concerned, most of the above cited studies only tested depolarized voltages (>−20 mV; [7,8,9,26]), and thus the increasing effect of pimozide at negative membrane potentials might have been missed. However, Seo et al. [25] did investigate the effect of pimozide at all membrane voltages and only found a blocking effect at positive membrane voltages. Taken together, the above studies are consistent with pimozide prolonging the overshoot (the positive phase) of the action potential in excitable cells. However, vestibular hair cells do not fire action potentials, and their receptor potential is not expected to turn positive because the reversal potential for the transducer current is ~0 mV. Therefore, in vivo, the major effect of pimozide on *I*_K,v_ expressed by vestibular type-II hair cells should be that of an increase in its amplitude and therefore hair cell membrane hyperpolarization.

As far as *I*_K,A_ is concerned, it was slightly blocked by pimozide, which is consistent with a previous study in rat hippocampal neurons showing a weak blocking effect of pimozide on *I*_K,A_ [9]. Since the block of *I*_K,A_ was quantitatively less important than the increase in *I*_K,v_, when both *I*_K,A_ and *I*_K,v_ were present in the same cell, the effect of pimozide on *I*_K,v_ predominated. In agreement with voltage-clamp results, current-clamp experiments showed that pimozide significantly hyperpolarized (tens of mV) the resting membrane potential of type-II hair cells and reduced the depolarization elicited by positive current steps (see Figure 10).

In vivo, hair cells hyperpolarization produced by pimozide would close voltage-gated Ca^2+^ channels expressed at the basolateral membrane [27,28], which are functionally coupled to spontaneous and evoked neurotransmitter (glutamate) exocytosis onto the afferent nerve terminal of vestibular primary sensory neurons (see [29] for a recent review). Therefore, our results are consistent with pimozide producing a decrease in neurotransmitter exocytosis due to hair cell hyperpolarization both at rest and during excitatory stimuli. Confirmation to this hypothesis could come from future studies comparing, before and after pimozide, the real-time changes in hair cell membrane capacitance as a measure of synaptic vesicles exocytosis produced by depolarizing voltage steps [30]. Another approach could be to test pimozide upon the afferent response recorded from the small afferent endings contacting type-II hair cells. However, the latter option is technically very demanding and to date has only been accomplished at the large calyceal afferent terminal enclosing type-I hair cells [31].

Noteworthy, pimozide was reported to block voltage-gated Ca^2+^ channels in the smooth muscles of rat *vasa deferentia* [4]. Although we did no*t* test for a possible effect of pimozide on Ca^2+^ channels in the present study, inhibition of glutamate exocytosis in vestibular type-II hair cells due to *I*_K,v_ increase could be reinforced by the block of Ca^2+^ channels.

Among the most common side effects of pimozide, there are dizziness and loss of balance [2]. It is tempting to speculate that these adverse effects are related to the effect of pimozide upon vestibular type-II hair cells, since the normal vestibular sensory input would be reduced. However, it should be mentioned that possible targets of pimozide could also be K^+^ channels expressed by type-I hair cells [32,33], even in consideration of the role of K^+^ in non-quantal transmission at the type-I hair cell-calyx afferent synapse [34], and/or by vestibular neurons.

On the other hand, in patients suffering from vestibular disorders of peripheral origin, the inhibition of the vestibular input might be useful. As many as 35% of adults aged 40 years or older have experienced some form of vestibular dysfunction [35]. Just as an example, Ménière disease affects 2 in 1000 people [36]. Whatever the pathological cause of chronic vestibular disease is, vertigo is the consequence of abnormal signaling from vestibular hair cells. Chemical or surgical ablation of vestibular hair cells must be used sometimes in order to alleviate the symptoms of unbearable vertigo: by killing vestibular hair cells, the brain no longer receives the wrong information that the head is rotating when it is actually not. Unfortunately, surgical or chemical ablation of vestibular hair cells often causes permanent hearing loss as a side effect. Irreversible hearing loss, in the case of surgery, occurs because vestibular and cochlear organs are strictly anatomically connected. On the other hand, the ototoxic drugs currently available, i.e., aminoglycoside antibiotics, kill both vestibular and cochlear hair cells. A drug that inhibits afferent vestibular transmission would be of great social benefit, especially since hearing deficit does not appear as a side effect of pimozide.

In any case, although during development avian and mammalian vestibular hair cells acquire a similar pattern of ion channels [37], a necessary future step will be to test pimozide on mammalian animal models. The latter may include electrophysiological recording from vestibular hair cells and their primary afferent neurons, as well as behavioural tests in healthy animals [38] or in animal models of vestibular disorders [39].

Outside the specific field of vestibular research, it is worth considering the broader implications of the effect of pimozide on K,v channels found here. More than 40 types of voltage-gated potassium (K,v) channels have been discovered so far, which are involved in diverse physiological processes, offering tremendous opportunities for the development of new drugs specific for several disorders (reviewed in [40]). Remarkably, the action of pimozide on K,_V_ channels reported here resembles that of retigabine on KCNQ/K_V_7 channels, since it similarly shifts the activation threshold towards more hyperpolarized voltages by ~15 mV and accelerates the activation kinetics [41]. In other words, as with retigabine, pimozide acts as a K^+^ channel opener. By reducing neuronal excitability, K_V_7 openers have a prominent role in the treatment of epilepsy [41]. The possibility that pimozide also acts as a K^+^ channel opener in mammalian excitable cells opens up new perspectives for its therapeutic use. Identification of the K,v subunit targeted by pimozide, e.g., by single-cell RT-PCR [42] or by the patch-seq technique, a novel approach that enables sequencing of total mRNA cell content after whole-cell recording (reviewed in [43]), will therefore be of great importance.

As a final remark, a better knowledge of the mechanisms of action of pimozide is also important since its use could already expand in the next few years. For example, recent studies demonstrated that pimozide, tested in genetic models of amyotrophic lateral sclerosis (ALS), can act as a neuroprotective compound, restoring neuromuscular transmission in *C. elegans*, zebrafish, and mouse models of ALS [44]. Additionally, pimozide significantly reduced the proliferation of brain cancer cell lines by inducing apoptosis [45].

## 5. Conclusions

Our study reveals a novel effect of pimozide as a K,_V_ channel opener. This finding provides a new key to understanding the beneficial and collateral pharmacological effects of this drug, as well as the basic knowledge for new potential therapeutic applications.

## Figures and Tables

**Figure 1 biomedicines-11-00488-f001:**
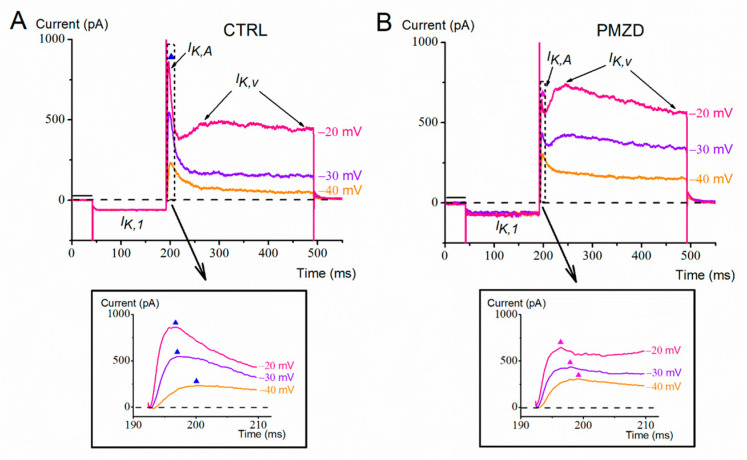
Representative whole-cell currents recorded from a type-II hair cell (E15). Here and in the next figures: CTRL = control condition; PMZD = pimozide; horizontal dashed line = zero-current level. (**A**,**B**), Membrane current responses recorded in control condition (**A**) or in the presence of pimozide [3 µM] (**B**) evoked by voltage steps to −40 mV, −30 mV and −20 mV after conditioning at −120 mV. The thick horizontal bar above the first region of the current indicates the holding current. The dashed rectangle indicates the time window of 17 ms considered for the expanded trace region shown in the panels at the bottom. The filled triangles refer to the peak of *I*_K,A_.

**Figure 2 biomedicines-11-00488-f002:**
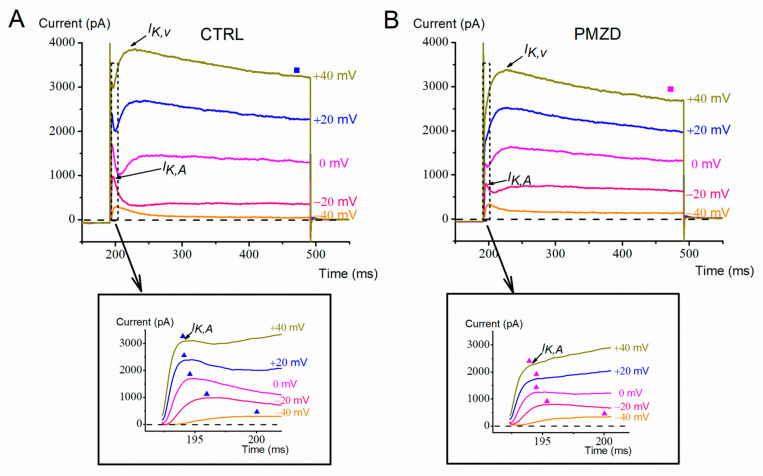
Representative whole-cell currents recorded from a type-II hair cell (E15). (**A**,**B**), Membrane current responses recorded in control condition (**A**) or in the presence of pimozide [3 µM] (**B**) evoked by voltage steps as indicated next to each trace, after conditioning at −120 mV (only the last part of the response to −120 mV is shown). The dashed rectangle indicates the time window (17 ms) considered for measuring the peak current. The corresponding expanded traces are shown below each panel. The filled triangles indicate the time points at which the peak current was measured. The steady-state current was measured towards the end of the depolarizing steps (filled squares).

**Figure 3 biomedicines-11-00488-f003:**
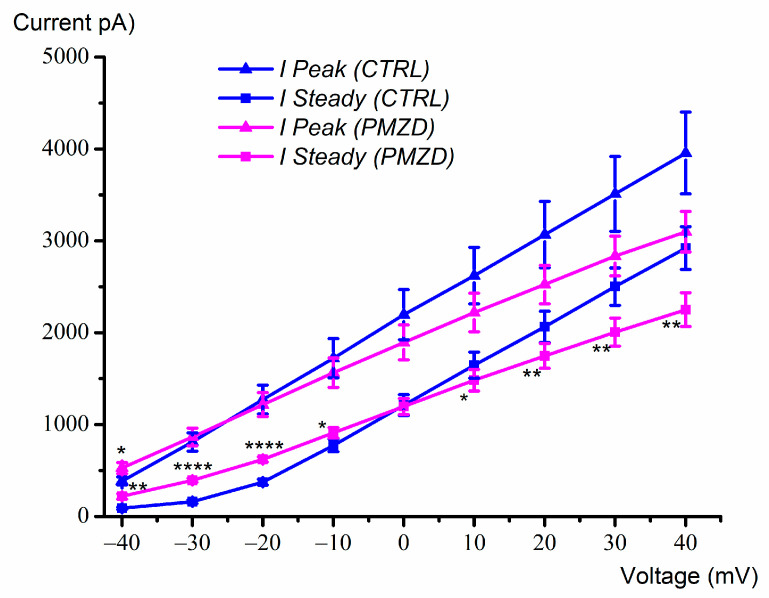
Average peak and steady-state current-voltage (*I–V*) curves obtained from type-II hair cells with and without pimozide (E15-E21; n = 7). Values are shown as mean ±S.E.; see Appendix A. * *p* ≤ 0.05; ** *p* ≤ 0.01; **** *p* ≤ 0.0001.

**Figure 4 biomedicines-11-00488-f004:**
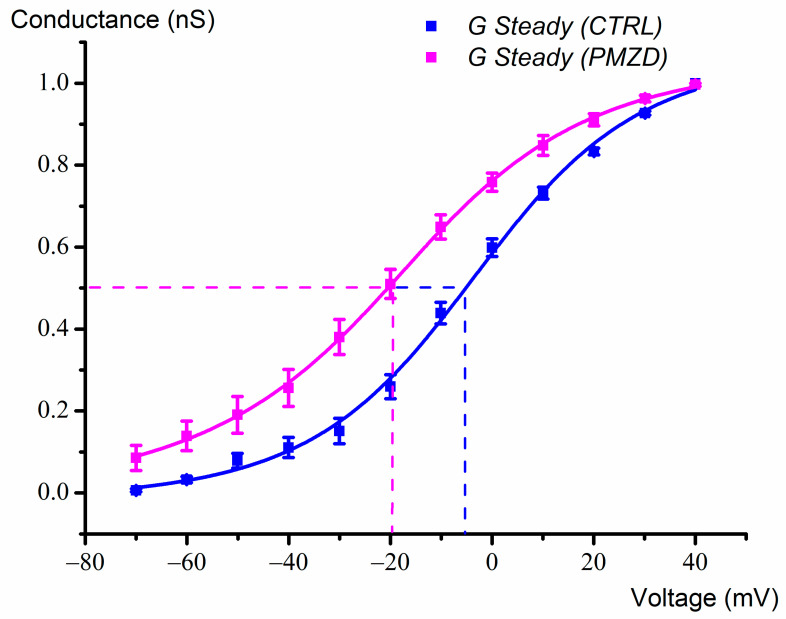
Normalized chord conductance/voltage relation (E15-E21; n = 7) (mean ±S.E.; see Appendix A) for *I*_K,v_. Fitting was performed with the Boltzmann function (see Methods). The half-activation voltage (*V*_1/2_) was −4 mV in control and −18 mV in pimozide (see dashed lines). The slope factor was 16 in control condition and 18 in pimozide.

**Figure 5 biomedicines-11-00488-f005:**
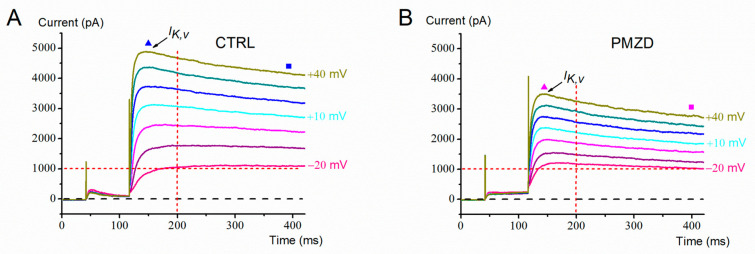
Representative whole-cell currents recorded from a type-II hair cell (E18). (**A**,**B**), Membrane current responses recorded in control condition (**A**) or in the presence of pimozide (**B**). In control condition, *I*_K,A_ was fully inactivated at the end of the 75 ms conditioning voltage of −40 mV. Note the change in the current response during the conditioning voltage of −40 mV in pimozide, consistent with the inhibition of *I*_K,A_ and the increase and acceleration of *I*_K,v_. The filled triangles indicate the time points at which the peak current was measured. The steady-state current was measured towards the end of the depolarizing steps (filled squares).

**Figure 6 biomedicines-11-00488-f006:**
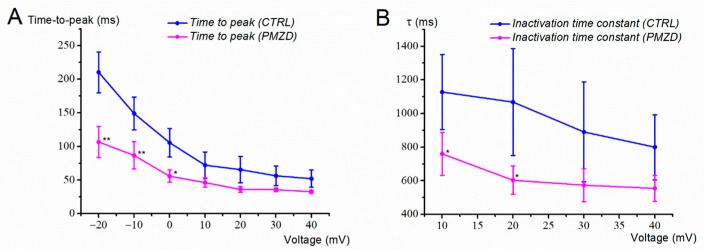
I_K,v_ kinetics as a function of membrane voltage. (**A**), time-to-peak. (**B**), inactivation time constant. The inactivation time constant was obtained by fitting the decaying portion of the current with Equation (3) (see Methods). Not all cells showed a clear inactivation during the 1 s voltage steps less depolarized than 10 mV in control conditions, which is why values are only shown from 10 mV. Data were obtained from 8 type-II hair cells (E15-E19) following I_K,A_ inactivation by the 75 ms conditioning voltage of −40 mV. Values are shown as mean ±S.E.; see Appendix A. * *p* ≤ 0.05; ** *p* ≤ 0.01.

**Figure 7 biomedicines-11-00488-f007:**
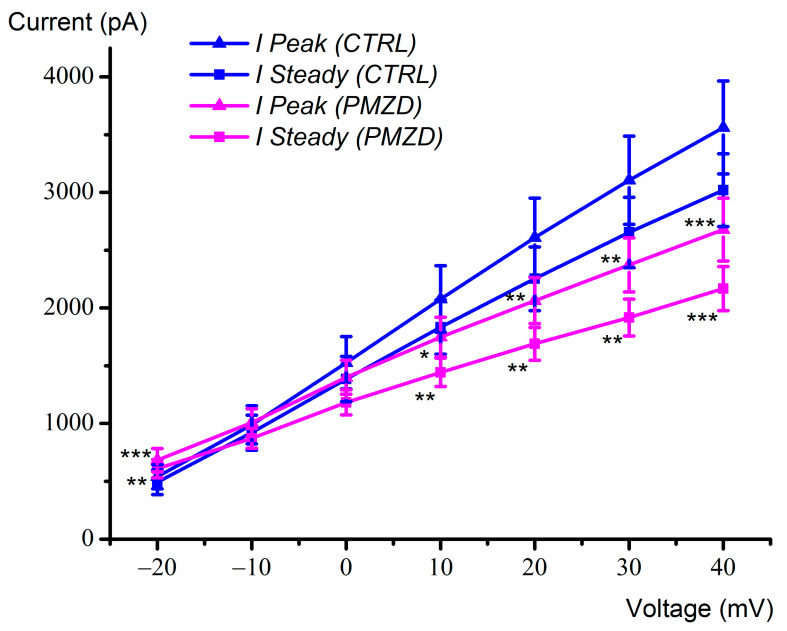
Peak and steady-state *I–V* relations after conditioning at −40 mV obtained from 8 type-II hair cells (E15-E19) with and without pimozide. Values are shown as mean ±S.E.; see Appendix A. * *p* ≤ 0.05; ** *p* ≤ 0.01; *** *p* ≤ 0.001.

**Figure 8 biomedicines-11-00488-f008:**
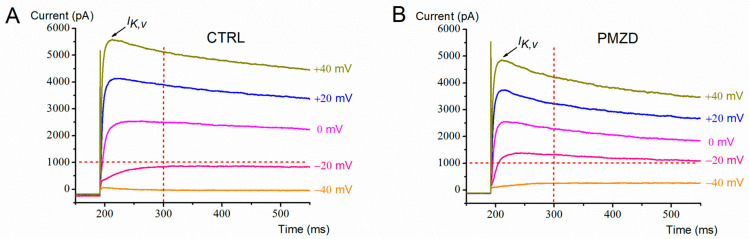
Representative whole-cell current recorded from an E10 type-II hair cell. (**A**,**B**), Membrane current response evoked by voltage steps as indicated next to each trace, after conditioning at −120 mV, in control condition (**A**) or in the presence of pimozide [3 µM] (**B**).

**Figure 9 biomedicines-11-00488-f009:**
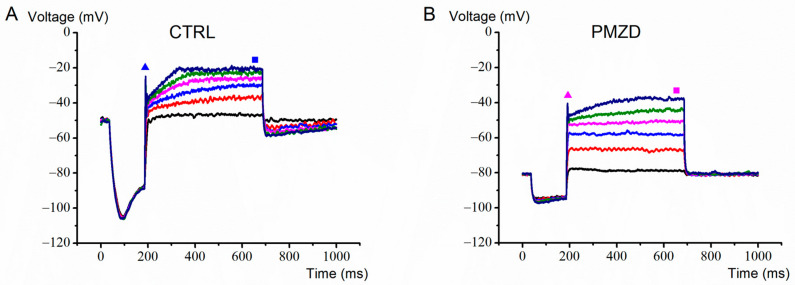
Representative voltage responses recorded from a type-II hair cell (E20) in control condition (**A**) or in the presence of pimozide (**B**). The cell was maintained at its resting membrane potential (zero current injected) and then hyperpolarized by a negative current step of −50 pA amplitude and 150 ms duration prior to iteratively applying depolarizing current steps of 500 ms in 50 pA increments starting from +10 pA. The filled triangles indicate the time points at which the peak voltage response was measured. The steady-state voltage response was measured at the end of the depolarizing current steps (filled squares).

**Figure 10 biomedicines-11-00488-f010:**
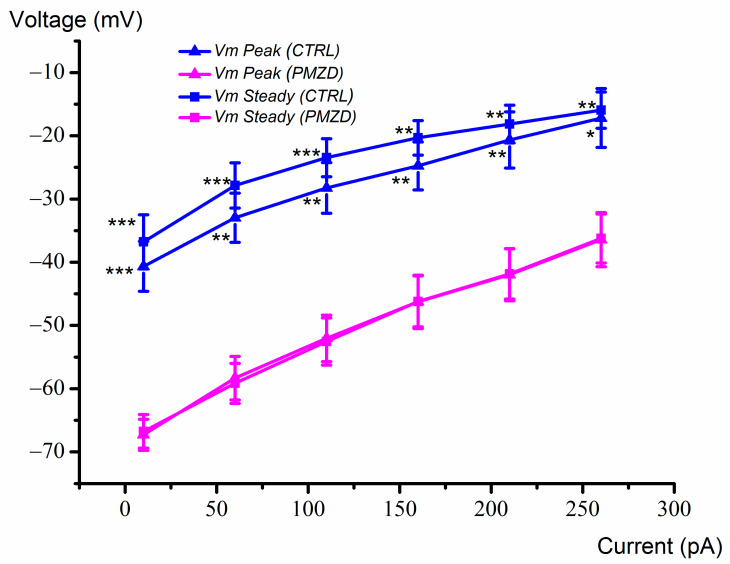
Average peak and steady-state voltage–current (V–I) relations. Values were obtained by measuring the peak and steady-state voltage response elicited by the current steps (n = 7; E15–E20), as shown by filled triangles and squares in Figure 9. Values are shown as mean ± S.E.; see Appendix A. * *p* ≤ 0.05; ** *p* ≤ 0.01; *** *p* ≤ 0.001.

**Figure 11 biomedicines-11-00488-f011:**
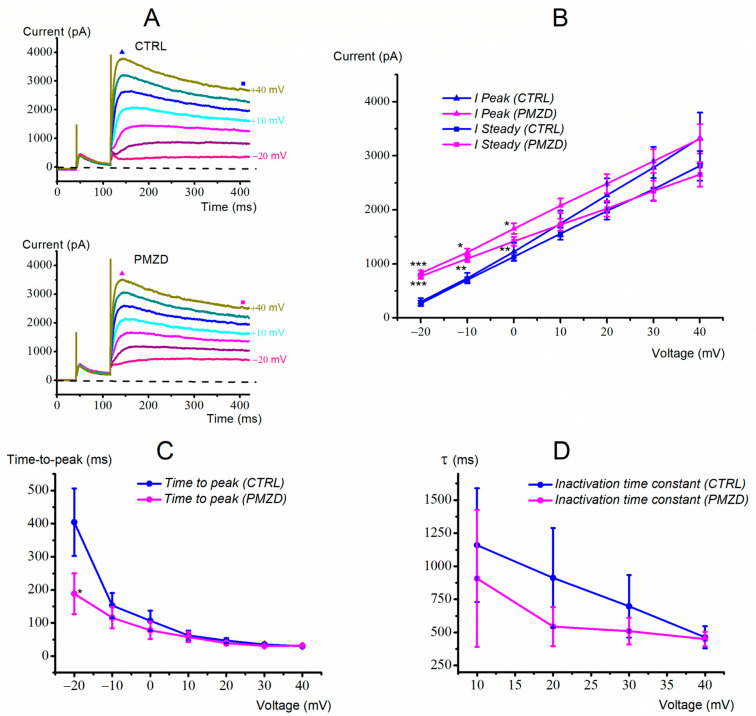
Voltage-clamp experiments with pimozide 0.3 μM. (**A**) Representative whole-cell current recorded from an E16 type-II hair cell in control condition (top panel) or in presence of pimozide (bottom panel). Conditioning voltage: −40 mV. (**B**) Average peak and steady-state *I–V* relations after conditioning at −40 mV, obtained from 7 type-II hair cells (E15-E21), in control condition or in the presence of pimozide. (**C**) Average time-to-peak in control condition or in the presence of pimozide (**D**), average inactivation time constant in control condition or in the presence of pimozide. All values are shown as mean ±S.E. See Appendix A. * *p* ≤ 0.05; ** *p* ≤ 0.01; *** *p* ≤ 0.001.

## Data Availability

Data can be made available upon direct request to the corresponding author.

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
