# Peer review of "Pimozide Increases a Delayed Rectifier K+ Conductance in Chicken Embryo Vestibular Hair Cells"

_biomedicines, 2023, doi:10.3390/biomedicines11020488_

Round 1

Reviewer 1 Report

The authors investigated the effect of pimozide on chicken embryo vestibular type-II hair cells K+ channels. They found that pimozide blocks a transient outward rectifying A-type K+ current but substantially increases a delayed outward rectifying K+ current. The net result was a significant hyperpolarization of type-II hair cells at rest and a strong reduction of their response to depolarizing stimuli. Their results are consistent with an inhibitory effect of pimozide on type-II hair cell synaptic transmission.

 ----------------

I was wondering of a different explanation for their results. This is that if pimozide blocks Ca current. Then Ca activated K current will not activate. Depending on the size relationship between the inward Ca current and the K current, block of the Ca current can be seen as a decrease in the outward current. In fact, it would be seen as a voltage dependent change in outward K current. With greatest effect at around -10 mV (which is about the maximum Ca current), and effect will be slower at depolarized potentials reaching about zero at -40 (about the reversal potential of the Ca current).

Question is whether or not, the Ca activated K current is expressed in type II hair cells at the age you selected for the study. A comment about this will be pertinent, also if IK,Ca would be implied in your results ?

In an ulterior characterization of pimizide action on K channels it may be interesting to determine the use dependency by means of repetitive pulse stimulation.

Reviewer 2 Report

The manuscript “Pimozide hyperpolarizes chicken embryo vestibular type-II hair cells” by Giunta et al is a research article which tested the effects of pimozide, a conventional antipsychotic drug, on K+ currents in chicken embryo vestibular type-II hair cells. The authors found that pimozide blocks a transient outward rectifying A-type K+ current while increasing a delayed outward rectifying K+ current. However, the net effect of pimozide was a significant hyperpolatization of type-II hair cells. These data suggest that pimozide could reduce synaptic transmission in type-II hair cells. Generally, the subject is of interest and scientifically sound and contains essential contents. This paper is also of importance for providing us the important evidence that pimozide causes an inhibitory effect on type-II hair cells. The manuscript has been well organized and written. However, I have several concerns on the paper.

1. To conclude that pimozide reduces synaptic transmission in type-II hair cells, the authors should show the effects of pimozide on spontaneous and evoked efferent synaptic events.

2. In this manuscript, statistical significance was assessed by student’s t-test. The authors should describe t values in the text.

3. Figure 3, 4, 6, 7, 10. Statistical significance should be assessed by ANOVA.

Reviewer 3 Report

Very interesting article that deserves an acceptance after a minor revision.

Two major points should be adressed:

1) The Autjors should provide a more detailed description of voltage-gated potassium channels in the cells. The channels should be named applying the IUPHAR (International Union of Pharmacology) nomenclature (see e.g. Gutman et al., 2005).

2) The Authors should explain why they did not perform the experiments for more concentrations of pimozide. The effect on the channels is likely to be pimozide concentration-dependent.

Minor comment:

1) Looking at the Figure 1 it was difficult for me to see that the holding current at -70 mV was really negative under control conditions. If it was true it would mean that this current was not carried by potassium ions (taking into account the Nernst equlibrium potential for this ion under physiological conditions).
